# The Loss of *focA* Gene Increases the Ability of *Salmonella* Enteritidis to Exit from Macrophages and Boosts Early Extraintestinal Spread for Systemic Infection in a Mouse Model

**DOI:** 10.3390/microorganisms10081557

**Published:** 2022-08-02

**Authors:** Ran Gao, Jian Zhang, Haoyu Geng, Yaonan Wang, Xilong Kang, Shizhong Geng, Xin’an Jiao, Paul Barrow

**Affiliations:** 1Jiangsu Co-Innovation Center for Prevention and Control of Important Animal Infectious Diseases and Zoonoses, Yangzhou University, Yangzhou 225009, China; 15893899527@163.com (R.G.); zj18705275313@163.com (J.Z.); g0201302022@163.com (H.G.); wangyaonan0915@163.com (Y.W.); xlkang@yzu.edu.cn (X.K.); jiao@yzu.edu.cn (X.J.); 2Jiangsu Key Laboratory of Zoonosis, Yangzhou University, Yangzhou 225009, China; 3Key Laboratory of Prevention and Control of Biological Hazard Factors (Animal Origin) for Agrifood Safety and Quality, Ministry of Agriculture of China, Yangzhou University, Yangzhou 225009, China; 4Joint International Research Laboratory of Agriculture and Agri-Product Safety of the Ministry of Education, Yangzhou University, Yangzhou 225009, China; 5School of Veterinary Medicine, University of Surrey, Daphne Jackson Road, Guildford GU2 7AL, UK; paul.barrow@surrey.ac.uk

**Keywords:** *Salmonella* Enteritidis, *focA* gene, exit from macrophages, extraintestinal spread, systemic infection, virulence

## Abstract

*Salmonella* Enteritidis (SE) can spread from the intestines to cause systemic infection, mainly involving macrophages. Intramacrophage *Salmonella* exits and reinfects neighboring cells, leading to severe disease. *Salmonella* genes involved in exiting from macrophages are not well understood or fully identified. A *focA::*Tn5 mutant was identified by an in vitro assay, with increased ability to exit from macrophages. A defined SEΔ*focA* mutant and its complemented derivative strain, SEΔ*focA*::*focA*, were constructed to confirm this phenotype. Although the lethal ability of *focA* mutants was similar to that of the parental SE in mice, it was isolated earlier from the liver and spleen than the parental SE. *focA* mutants induced higher levels of proinflammatory IL-12 and TNF-α compared with the parental SE and SEΔ*focA*::*focA*. *focA* mutants showed higher cytotoxicity and lower formate concentrations than SE and SEΔ*focA*::*focA*, whereas there was no change in pyroptosis, apoptosis and flagella formation ability. These current data suggest that the *focA* gene plays an important role in regulating intramacrophage *Salmonella* exiting and extraintestinal spread in mice, although the specific mechanism requires further in-depth studies.

## 1. Introduction

*Salmonella**enterica* spp. remain a major foodborne zoonotic pathogen causing serious public health problems and economic losses [1,2]. During systemic infection in the host, *Salmonella* infects intestinal epithelial cells after entering the gut, where it proliferates and induces inflammation. This is followed by extracellular [3] and intracellular dissemination [4,5,6,7], the latter generally occurring in macrophages, where multiplication takes place. Transmission between macrophages is clearly important to spread infection within organs, such as the lymph nodes, spleen and liver, where further multiplication appears to take place. Multiplication within macrophages involves accumulation of relatively small numbers of bacteria before dissemination to adjacent susceptible macrophages occurs [8].

Exiting macrophages is as important as invasion for pathogens, which cycle between intra- and extracellular stages [9]. The exit phase is well understood for pathogens such as *Shigella* and *Listeria*, which exist predominantly within the cell cytoplasm, and intercellular transfer occurs via actin-mediated protrusions projecting into neighboring cells [10,11]. Other mechanisms expressed by pathogens, such as *Chlamydia* [12] and *Legionella* [13], have also been described.

Although the abilities of invasion and replication within cells have been the major focus of research on *Salmonella* pathogenicity, the ability of intramacrophage *Salmonella* exiting has received little attention. The means by which bacteria exit the cell and whether this involves some form of cell death [14] remains unclear. It is well known that *Salmonella* has the capacity to induce cell death at different times after infection [15,16]. The SipB protein induces rapid cell death through activation of caspase-1, with fragmentation of chromatin and cytoplasmic membrane blebbing [17,18,19]. Pyroptosis is also caspase-1-dependent. In contrast, apoptosis is known to be caspase-3-dependent [20]. The cell swelling that precedes necrosis and bacterial release induced by motile *Salmonella* is thought to be flagella-related [21].

Whether genes such as *sifA* are involved is unclear, as *sifA* is involved information of the *Salmonella*-containing vacuole (SCV), preventing *Salmonella* from entering the cytoplasm; *sifA* mutants diminish the integrity of the SCV [22]. The *prgJ* gene may also be involved in the exit process [23], possibly due to its involvement in pyroptosis caused by *Salmonella* also involving flagella, although in this case, it is thought to be a defense mechanism in pig lymph nodes [24].

Because very few bacterial genes have been identified in relation to *Salmonella* exiting from macrophages, we decided to screen a mini-Tn*5* transposon mutant library of *Salmonella* Enteritidis C50041 to identify genes required for *Salmonella* to exit from macrophages. Derivatives of the mini-Tn*5* transposon-carrying selectable antibiotic resistance markers are powerful tools for mining bacterial genes related to phenotypes and functions [25]. We found that the *focA* gene [26], expressed as the formate transporter FocA, is involved in regulating the ability of *Salmonella* to exit from macrophages, as its loss quantitatively increased the exiting ability, which boosted early extraintestinal spread for systemic infection in mice.

## 2. Materials and Methods

### 2.1. Bacterial Strains, Plasmids and Cells

The bacterial strains, plasmids and cells used in this study are listed in Table A1. The defined bacterial deletion mutant was produced according the method of scarless–markerless genome genetic modification [27].

### 2.2. Mice and Animal Ethics

Specific pathogen-free (SPF) female BALB/c mice (8 week; 20 ± 2 g) were obtained from the Comparative Medical Center of Yangzhou University (Yangzhou, China). All animal experiments were approved by the Animal Welfare and Ethics Committees of Yangzhou University and complied with the guidelines of the Institutional Administrative Committee and Ethics Committee of Laboratory Animals.

### 2.3. Construction of Tn5 Mutant Library and Mutant Screen

SE C50041 was used for random transposon mutagenesis by a mini-Tn*5* transposon delivered on suicide vector pUT with a kanamycin-resistant gene, as described in [28]. A Tn*5* mutant library was constructed by conjugating *E. coli* χ7213 (mini-Tn*5*) as donor strain with C50041, which is sensitive to kanamycin, as recipient. The transconjugants were isolated on LB agar containing 50 mg/mL chloramphenicol and 100 μg/mL kanamycin.

The mutant screen was performed as described previously. Briefly, RAW264.7 cells (5.0 × 10^5^ cells/well) were cultured in Dulbecco’s Modified Eagle Medium (DMEM) containing 10% fetal bovine serum (GenDEPOT Inc, Barker, TX, USA) for 12 h at 37 °C in 24-well plates. Mutants (MOI = 100:1) were added to the culture medium, and the plates were centrifuged at 1000 rpm for 10 min for *Salmonella* to be deposited onto the surface of RAW264.7 cell s. The cells were then incubated at 37 °C for 1 h, washed twice with sterile phosphate-buffered saline (PBS) and incubated in DMEM plus 100 mg/mL gentamicin (GM, Sigma Aldrich, St. Louis, MO, USA) for 1 h. The culture medium was changed to DMEM with 10 mg/mL gentamicin. After 8 h, the medium with gentamicin was removed, and new DMEM without antibiotics was added for another 1 h. The cells were lysed with 0.2% Triton X-100 for 10 min at 37 °C. The loads of *Salmonella* in the culture medium and inside cells were counted, and their ratios were calculated and compared.

### 2.4. Identification of Sequence Flanking Tn5 Inserted in Bacterial Genome

The sequence-flanking Tn*5* inserted in the bacterial genomes was amplified by PCR [27]. The primers are listed in Table A2. Briefly, bacterial genomic DNA was isolated from mutants and digested with *Nla* III (New England Biolabs, Hitchin, Herts, UK). An adaptor of double-stranded DNA was ligated to the genomic DNA, a special PCR was performed once with primer set Y linker/P6U and twice with Y linker/Tn5-p, and the PCR product was sequenced by a Tn*5*-p primer. Homology searches were performed using the public databases BLASTn and BLASTx at http://www.ncbi.nlm.nih.gov, accessed on 6 July 2020.

### 2.5. Cosntruction of SEΔfocA and SEΔfocA::focA

According the protocol based on pGMB152 suicide plasmid [28], SEΔfocA was constructed by the chloramphenicol resistance gene replacing the *focA* gene, and the complemented strain *SEΔfocA::focA* was generated with pBR322-*focA* using a method described in [29].

### 2.6. In Vitro Exiting Ability of Intramacrophage Salmonella focA Mutants

The assay was performed as the *Salmonella* mutant screen described in Section 2.3.

### 2.7. Virulence Analysis of Salmonella focA Mutant in Mice

#### 2.7.1. Extraintestinal Spread

The capacity for the extraintestinal spread of *Salmonella* mutants was analyzed by monitoring *Salmonella* loads in the murine liver and spleen. Mice were infected orally with 1.0 × 10^7^ CFU in 100 μL of each SE strain/mutant. Three mice in each group were euthanized at 1, 4 and 7 dpi, and a section of the liver and spleen were removed aseptically, weighed and homogenized individually in 1 mL PBS. Dilutions of the homogenates (100 μL each) were plated on XLT4 agar and incubated at 37 °C overnight. Bacterial colonies were counted and expressed as Log10 CFU/g, with negative samples reported as 0 CFU/g.

#### 2.7.2. Lethal Ability

The lethal ability of the *focA* mutant for mice was performed as described in [30]. Mice were inoculated orally with 0.2 mL of *Salmonella* mutant (approximately 2.0 × 10^6^ CFU). The mouse survival rate of each group (n = 10, 5 group) was calculated after two weeks.

##### 2.7.3. mRNA Level of Cytokines in Murine Spleen

The spleens of mice infected with *Salmonella* were collected on post-infection days 1, 4 and 7, and total RNA was extracted using an RNeasy mini kit (Qiagen, Valencia, CA, USA). cDNA synthesis was carried out using reverse transcriptase PCR (Master cycler, Eppendorf, Hamburg, Germany). cDNA was diluted 5 times as a template for qRT-PCR, and the SYBR method was used to detect the mRNA level of cytokines with the primers listed in Table A3.

### 2.8. Biological Feature Analysis for Possible Mechanisms

#### 2.8.1. Formate Level in the Bacteria

Formate concentrations were measured by a formate assay kit (Abcam, Cambridge, MA, USA). The formate level was detected in the bacteria at 3, 5 and 9 h.

With the Abcam formate assay kit, formate is oxidized to generate a product resulting in color formation (λ = 450 nm) proportional to the formate concentration. In brief, formate assay buffer, enzyme mix and substrate mix were added to each standard and test sample in 96-well plates and incubated for 60 min at 37 °C.

The OD_450nm_ was measured in a microplate reader proportional to the formate concentration [31].

#### 2.8.2. Formate Level in *Salmonella*-Infected Macrophages

The steps for *Salmonella* infection to macrophage were the same as those described Section 2.3. After 9 h, the culture medium with 10 mg/mL gentamicin was removed, washed twice with sterile PBS and lysed in formate assay buffer at a ratio of 1.0 × 10^6^ cells per 100 μL buffer. Lysates were transferred into Eppendorf tubes and centrifuged for 10 min at 14,000 rpm. The supernatant was used to detect the formate level with a formate assay kit (Abcam, USA) [31].

#### 2.8.3. LDH Assay for Cytotoxicity

The cytotoxicity of the *focA* mutants was evaluated by LDH release from *Salmonella*-infected cells. Cell culture and bacterial infection were performed as in the mutant screen described above. DMEM with 10 μg/mL gentamicin was added for 3 h, and the LDH level released in the cell medium was detected by an LDH cytotoxicity assay detection kit (Beyotime, Nantong, China).

#### 2.8.4. Pyroptosis and Apoptosis Assessment for Cell Death

*Analysis of pyroptosis*: J774A.1 cells were seeded into 12-well plates at a density of 5.0 × 10^5^ cells per well and infected with *Salmonella* as described above. After harvesting the supernatants, the remaining cells were lysed directly with 300 μL cell lysis buffer per well, and the supernatant and lysate from each well were mixed. The mixtures were centrifuged at 2000 rpm for 5 min to remove cell debris. An equal volume of methanol and a 0.25 volume of chloroform were added, vortexed vigorously and centrifuged at 12,000 rpm for 5 min. The supernatant was aspirated completely. An equal volume of methanol was added to each sample, vortexed vigorously and centrifuged at 12,000 rpm for 5 min. The protein pellets were dried at 55 °C for 10 min, resuspended with 40 μL of 1 × SDS-PAGE sample-loading buffer (Beyotime, China) and boiled for 10 min at 95 °C. The samples were loaded onto 15% Tris-glycine gels and analyzed by Western blot [32]. The primary antibody used in this study was anti-caspase-1 p10 antibody (AG-20B-0042-C100, AdipoGen, San Diego, CA, USA). The secondary antibodies were goat anti-mouse IgG-HRP.

*Analysis of apoptosis*: An annexin V-FITC/PI double staining method was used to detect apoptosis. After RAW264.7 cells were infected with SE and incubated in DMEM plus 10 mg/mL gentamicin for 3 h, the cells were stained with an annexin V-FITC kit (Miltenyi) and analyzed by flow cytometry. The specific operation was as follows: 1.0 × 10^6^ cells were collected and washed in 1 mL of 1 × binding buffer and centrifuged at 12,000 rpm for 10 min. After the supernatant was removed completely, cells were resuspended in 100 µL of 1× binding buffer and 10 µL of annexin V-FITC and incubated for 15 min in the dark at room temperature. Cells were washed again by adding 1 mL of 1× binding buffer and centrifuged at 12,000 rpm for 10 min. Supernatant was removed completely, and cells were resuspended in 500 µL of 1× binding buffer. After 5 µL of PI solution was added, cells were immediately analyzed by flow cytometry [33].

#### 2.8.5. Motility Analysis for Flagella

Bacterial motility was analyzed by U tube and semisolid agar plate.

*U tube:* Fresh cultured single colonies of the SE strains were selected from a solid LB plate and used to inoculate one side of a U tube containing 6 mL semisolid LB medium. The U tube was incubated at 37 °C for 8 h, and the growth of the bacteria was observed from the other side of the U tube.

*Semisolid agar plate:* SE strains were cultured in LB liquid broth to their logarithmic phase. After washing the bacterial cells twice with PBS, the bacterial density was adjusted to OD600 nm = 1.0. A freshly prepared semisolid LB plate containing 0.5% agar was used to detect motility. A volume of 10 μL of the bacterial suspension was pipetted to the center of the semisolid plate. The plate was allowed to dry for 20 min and incubated at 37 °C for 20 h. Motility was evaluated by the diameter of the visible bacterial growth.

*Electron microscope: Salmonella focA* mutant cultures were negatively stained with 0.1% phosphotungstic acid solution for 1 min. The flagella were observed under an electron microscope.

### 2.9. Statistical Analysis

The bacterial CFUs, mouse survival and morphometric analysis data were analyzed using GraphPad Prism 7 (GraphPad Software, LaJolla, CA, USA). Analysis of variance (ANOVA) was performed to compare the mutant groups with the C50041 control, as well as to compare the mutant groups with the PBS control. All results are expressed as the mean ± SEM. Statistical significance was assigned at *p* values <0.05 (*), <0.01 (**) or <0.001 (***) based on a Student’s *t*-test.

## 3. Results

### 3.1. focA::Tn5 Mutant with Imrpoved Exiting Ability from Macrophages

A total of 887 conjugants were screened from the Tn*5* mutant library of C50041, and one mutant showed improved exiting ability from the RAW264.7 macrophages. Following amplification of the Tn*5*-flanking sequence by PCR and BLAST analysis, the Tn5-inserted gene was identified as *focA*.

### 3.2. SEΔfocA Mutant Reconfirmed

To confirm that the exiting ability of the SE *focA* mutant improved, the SEΔ*focA*-deletant and SEΔ*focA::focA-*complemented strain were constructed. RAW264.7 cells were infected with *focA::*Tn5, SEΔ*focA*, SEΔ*focA::focA* and C50041, and their exiting abilities from RAW264.7 cells were analyzed. As shown in Figure 1, 1 h after removal of antibiotics, the numbers of SEΔ*focA* in the culture medium increased significantly compared to C50041 (*p* < 0.05), whereas the intracellular numbers were very similar. The ratio of extracellular/intracellular bacterial count of SEΔ*focA* was also increased significantly (*p* < 0.05) compared to that of C50041. The exiting ability of SEΔ*focA* was similar that of *focA::*Tn5 (*p* < 0.01), and the exiting ability of SEΔ*focA::focA* was similar to that of C50041 (*p* > 0.05).

### 3.3. Virulence Analysis of Salmonella focA Mutants in Mice

#### 3.3.1. Improved Early Extraintestinal Spreading Ability

After 1.0 × 10^7^ CFU in 100 μL *focA::*Tn*5*, SEΔ*focA*, SEΔ*focA::focA* and C50041 were administered orally to mice. The in vivo dynamics of the four SE strains showed that *focA* mutants could be isolated from the livers and spleens one day after inoculation, whereas C50041 was not isolated at this time, and higher loads of *focA* mutants could be isolated from the liver and spleen compared to C50041 (Figure 2).

#### 3.3.2. No Obvious Change in Lethal Ability in Mice

The survival rates of the four SE strains were compared after oral inoculation of mice with 2.0 × 10^6^ CFU. The survival of mice infected with the *focA* mutants did not differ significantly from that of the mice infected with C50041 (Figure 3).

#### 3.3.3. Increased Ability to Promote Murine Proinflammatory Cytokines

The mRNA levels of the proinflammatory cytokines IL-12 (*focA::*Tn*5*: *p* < 0.05 and Δ*focA*: *p* < 0.05) and TNF-a (*focA::*Tn*5*, *p* < 0.05 and Δ*focA*, *p* < 0.05) exceeded those induced by C50041 at 1 dpi (Figure 4).

### 3.4. Biological Phenotype Analysis of focA Mutant for Possible Mechanisms

#### 3.4.1. Less Formate Produced by *focA* Mutant

*focA* mutants were deficient in formate production, as shown in Figure 5 (*p* < 0.05).

#### 3.4.2. Less Formate in *focA* Mutant-Infected RAW264.7

According to analysis by formate assay kit, the formate concentration in *focA* mutant-infected RAW264.7 cells was (*p* < 0.05) significantly lower than that in C50041-infected RAW264.7 cells, as shown in Figure 6.

#### 3.4.3. Increased Cytotoxicity by *focA* Mutants

The cytotoxicity of *Salmonella* was analyzed by LDH level in the culture medium of infected cells. As shown in Figure 7, LDH levels induced by *focA*::Tn*5* (*p* < 0.05) and Δ*focA* (*p* < 0.05) were significantly higher than those induced by C50041. LDH levels induced by SEΔ*focA*::*focA* and C50041 were similar.

#### 3.4.4. No Obvious Change in Cellular Pyroptosis Based on Caspase-1 Protein Measurement by *focA* Mutants

Western blotting was used to detect caspase-1 expression in J774A.1 macrophage-like cells after infection with the SE strains. The results are shown in Figure 8. Compared with the C50041 group, expression of P45 (Procaspase-1), caspase-1 and GAPDH in the SEΔ*focA* group was not altered significantly, suggesting that the mutation of the *focA* gene had no obvious effect on the ability of SE to cause pyroptosis.

#### 3.4.5. No Obvious Change in Cellular Apoptosis Based on Annexin V-FITC/PI Staining by *focA* Mutants

Apoptosis of infected RAW264.7 cells was detected by flow cytometry. The results are shown in Figure 9. Compared to the C50041 group (4.5% (early), 51.9% (late)), the apoptosis rate induced by *focA*::Tn*5*, Δ*focA* and Δ*focA*::*focA* did not change significantly (2.3% (early), 46.9% (late) for *focA*::Tn*5*; 3.1% (early), 46.1% (late) for Δ*focA*; and 3.6% (early), 45.3% (late) for Δ*focA*::*focA*). The results indicate that the *focA* gene is not involved in apoptosis induced by SE.

#### 3.4.6. No Obvious Change of Flagella Based on Bacterial Motility by *focA* Mutants

U tubes and semisolid agar plates were used to detect the motility of the SE *focA* mutants (Figure 10). U tubes showed no alterations in motility. In the semisolid agar plate, the diameter of the bacterial colony of SEΔ*focA* was 54 ± 2 mm, and that of C50041 was 61 ± 2 mm. Statistical analysis (Figure 10) showed that the size of the bacterial colony did not differ significant (*p* > 0.05), proving that the *focA* gene has no significant effect on SE motility. Normal flagella around the *focA* mutants were observed by electron microscopy.

## 4. Discussion

The ability of virulent *Salmonella* organisms to exit from infected macrophages, by whatever means, is clearly a key step in the process of bacterial spread from one susceptible cell to another, as occurs with some other intracellular pathogens [9,10,11,12]. This is also indicated by the fact that the number of bacteria within infected macrophages remains low and infection spreads to more susceptible macrophages [4]. In healthy individuals, the host can recognize and eliminate pathogens through innate and acquired immunity. *Salmonella* clearly face the challenge of immune mediators during their escape from the intracellular environment. Once in macrophages, invasive *Salmonella* are able to evade immune surveillance by manipulation of the intracellular environment, including the SCV, using sophisticated strategies [5,6,7,8].

Several *S.* Typhimurium genes have been identified, the mutations of which have been shown to reduce the ability to exit from macrophages and which also attenuate virulence. These genes include SipB, which induces rapid cell death through activation of caspase-1 [20], flagella [22] and possibly also SifA and PrgJ, although the case for these is less clear [14,23]

Extracellular escape is affected by many factors and, if it involves cell death, may take many forms [15,16,17].

We identified a mutation of *focA* in *S*. enteritidis, which increased the ability to exit from macrophages, suggesting that the FocA protein plays a key role in suppressing premature release of *Salmonella* bacteria. Given that the number of bacterial cells that accumulate in macrophages before release is relatively low, the dynamics of multiplication, together with release and exiting, could form an important study that could shed light on its role in this stage of the infection process. The means by which this protein is involved in exiting remain unclear. It is an important component in mixed acid fermentation and in metabolic switching, depending on carbon source and redox level [26,31].

In addition to the phenotype of the initial transposon mutant, we demonstrated that the *focA* mutant colonized the liver and spleen more rapidly following oral inoculation of mice in the early stages of infection, although it did not affect the bacterial load in these organs at 4 and 7 days post infection. As a result of this latter observation, the lethality of the *focA* mutant for mice was also not changed from that of the current parent strain. The results were obtained initially with a Tn*5* insertional mutation and confirmed by producing a defined mutant. In addition, a *focA*-complemented strain was constructed using the pBR322-*focA* construct. The phenotype of the complemented strain was very similar to that of the parent strain. A role of the *f**ocA* gene in negative regulation of the escaping ability of *Salmonella* Enteritidis in macrophages must be considered highly likely.

Attempts were made to ascertain the basic mechanism behind this phenotype. Key proinflammatory cytokines IL-12 and TNF-α were measured in vitro. The *focA* mutation led to increases in IL-12 and TNF-α, indicating that the FocA protein plays a role during infection in suppressing these early indicators of the immune response. Small changes were observed in the levels of IFNγ and IL-1β, but the differences were not statistically significant. Early indicators of innate immunity, including suppression of proinflammatory chemokines, are a key characteristic of typhoid (acute systemic disease)-producing *Salmonella* serovars invading from the intestine [34]. This is facilitated by the absence of flagella in serovars such as *S**almonella* Gallinarum and *S**almonella* Pullorum and suppression of flagellation and other virulence genes in *S**almonella* Typhi [35]. Thus, suppression of components of the early innate response may facilitate bacterial survival during transfer between susceptible macrophages.

The non-functional formate transport activity of the *focA* mutant [36] coupled with more rapid exiting capacity and cytotoxicity suggests that FocA is actively involved in regulating bacterial virulence in the early stages of systemic infection, possibly by or including suppression of inflammatory host signals. It has been reported that quantities of formate are secreted by *E. coli* and *Salmonella* during stationary-phase growth and that this contributes to increased resistance to antimicrobial peptides by virtue of their oxidation via the respiratory chain bypassing the site of inhibition between NADH dehydrogenase and quinone [37]. There is also an interesting relationship between formate accumulation in epithelial cells and *Shigella* virulence, with a mutation in pyruvate-format lyase reducing plaque production and virulence, which is restored by the addition of formate [38]. Koestler et al. also believe that intracellular formate is a signal for modulation of bacterial virulence factors.

Although the mechanism of *Salmonella* exiting from the intramacrophage environment currently remains unclear, programmed cell death, including pyroptosis, does not seem to be involved. Intracellular *Salmonella* activates the NLRC4 inflammasome, mainly through the *Salmonella* pathogenicity island-1 type III secretion system (T3SS) and flagella, which further activates caspase-1 and causes pyroptosis [39,40]. Apoptosis could be activated by *Salmonella* pathogenicity island-1 effectors through activated caspase-3-induced pathways, including both intrinsic and extrinsic pathways in *Salmonella*-infected macrophages [41], and it could be a possible strategy for induced intracellular *Salmonella* bacteria to exit from macrophages. Although we found that *focA* mutants induced increased cytotoxicity, further analysis by Western blot and flow cytometry showed that there was no change in induction of pyroptosis and apoptosis. Although flagella may facilitate bacterial escape from macrophages [22], in this study, we observed no change in motility or physical appearance by electron microscopy.

Our preliminary results show that the *focA* gene plays a significant role in the ability of intracellular *Salmonella* to exit from macrophages in vitro and increases early-stage extraintestinal spread in systemic infection without affecting lethal ability. The nature of the interaction between the metabolic function of FocA and its contribution to the early stages of systemic infection clearly require further investigation.

## 5. Conclusions

In this study, a *focA* mutation was identified from a mini-Tn*5* transposon mutant library of *S.* enteritidis C50041, which displayed stronger exiting ability from macrophages and boosted early extraintestinal spread in mice. This result indicates that the *focA* gene negatively regulated the *S*. enteritidis exiting ability from macrophages, although this mechanism requires further in-depth studies.

## Figures and Tables

**Figure 1 microorganisms-10-01557-f001:**
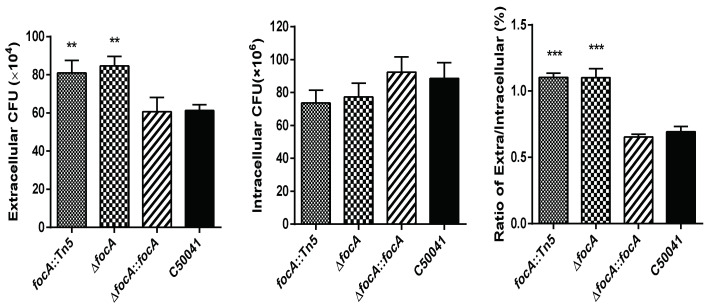
Exiting ability of intracellular SE *focA* mutants from RAW264.7. Data are presented as mean ± SEM of three independent experiments; ** *p* < 0.01, *** *p* < 0.001.

**Figure 2 microorganisms-10-01557-f002:**
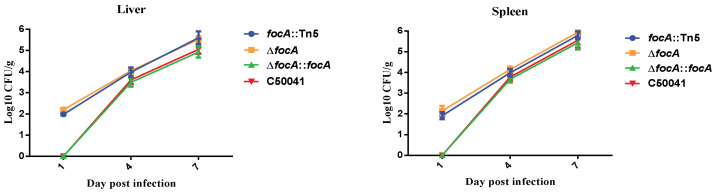
Extraintestinal spread of SE *focA* mutants by bacterial loads in murine liver and spleen. Data are presented as mean ± SEM of three independent experiments.

**Figure 3 microorganisms-10-01557-f003:**
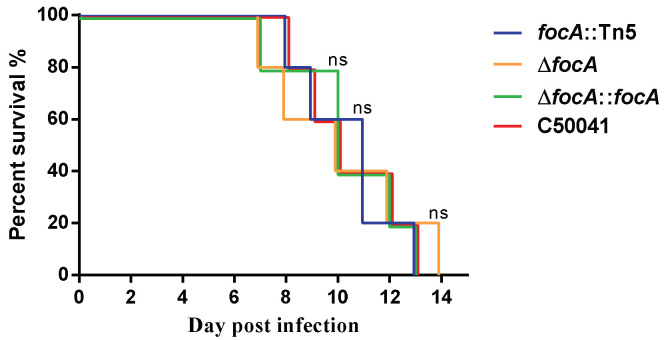
The survival of mice infected by SE *focA* mutants. Data are presented as mean ± SEM of three independent experiments.

**Figure 4 microorganisms-10-01557-f004:**
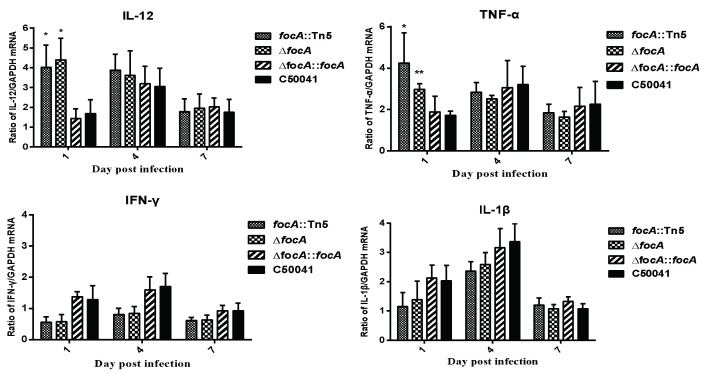
mRNA levels of cytokines in murine spleen caused by SE *focA* mutants. Data are presented as mean ± SEM of three independent experiments; * *p* < 0.05, ** *p* < 0.01.

**Figure 5 microorganisms-10-01557-f005:**
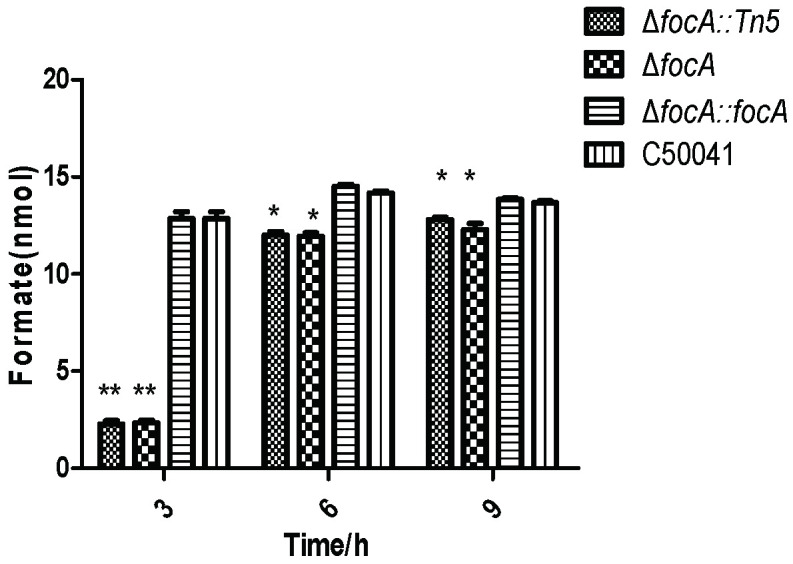
Formate concentration in SE *focA* mutant at different culture times. Data are presented as mean ± SEM of three independent experiments; * *p* < 0.05, ** *p* < 0.01.

**Figure 6 microorganisms-10-01557-f006:**
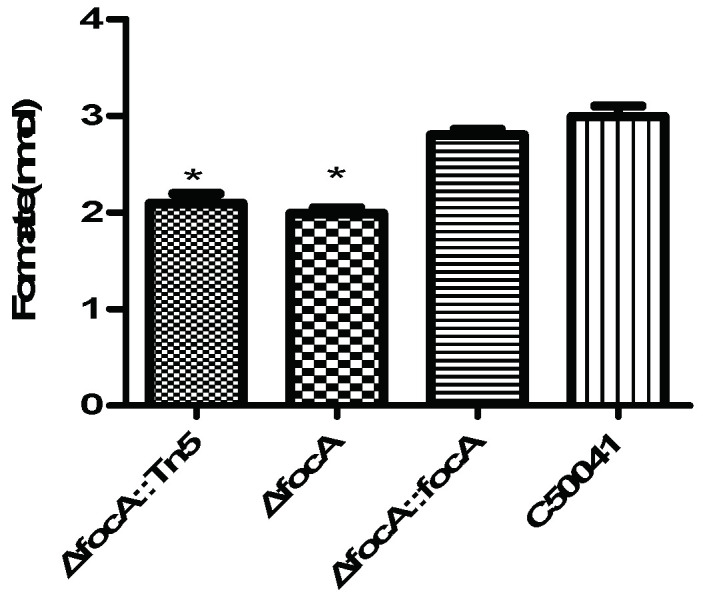
Formate concentration in *Salmonella*-infected RAW264.7 (MOI = 100, T = 9). Data are presented as mean ± SEM of three independent experiments; * *p* < 0.05.

**Figure 7 microorganisms-10-01557-f007:**
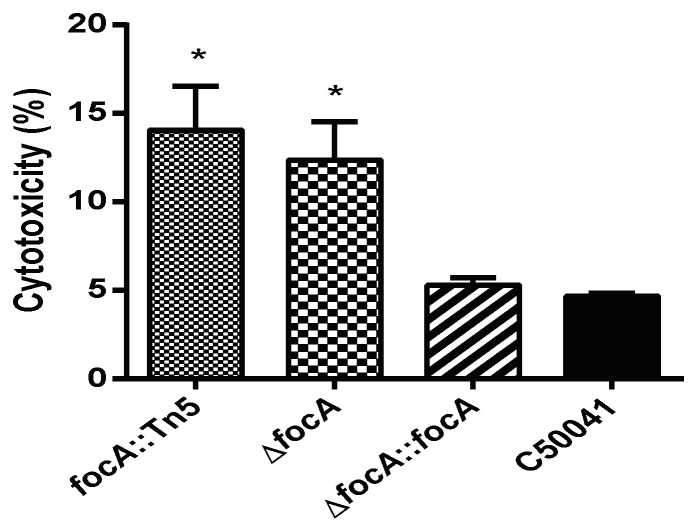
The cytotoxicity of SE *focA* mutants to RAW264.7. Data are presented as mean ± SEM of three independent experiments; * *p* < 0.05.

**Figure 8 microorganisms-10-01557-f008:**
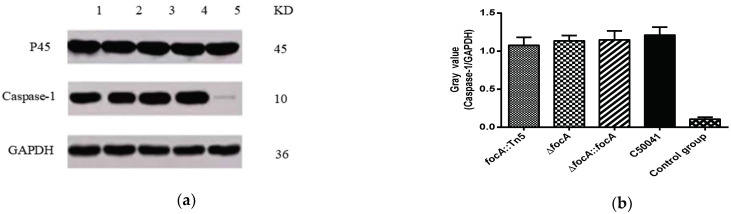
The ability of SE *focA* mutants to induce expression of caspase-1 protein: (**a**) Western blot analysis of caspase-1 protein: 1: *focA::*Tn*5*, 2: Δ*focA*, 3: Δ*focA::focA*, 4: C50041, 5: control group; (**b**) gray analysis of caspase-1 protein. Data are presented as mean ± SEM of three independent experiments.

**Figure 9 microorganisms-10-01557-f009:**
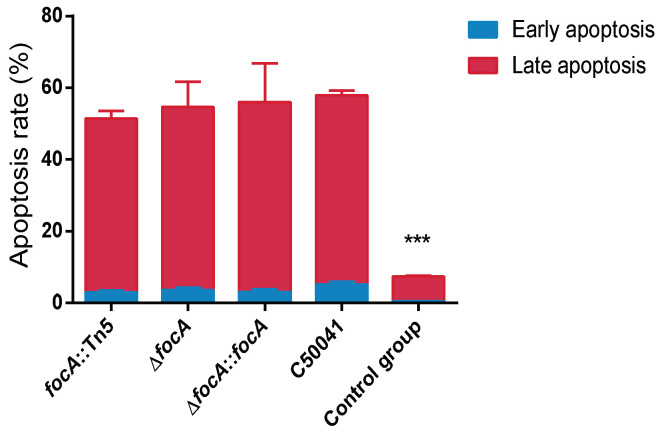
The apoptosis of the *Salmonella*-infected RAW264.7 was detected by flow cytometry. Data are presented as mean ± SEM of three independent experiments. *** *p* < 0.001.

**Figure 10 microorganisms-10-01557-f010:**
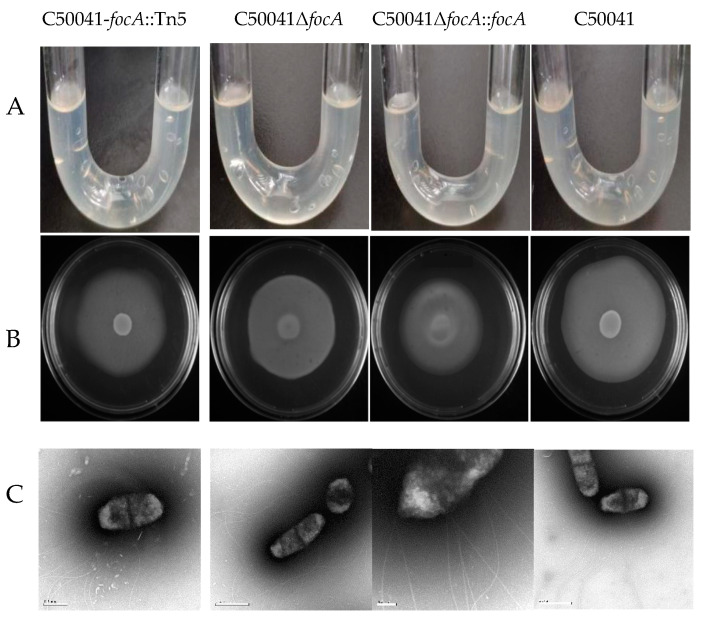
Motility analysis and flagellar observation of *Salmonella focA* mutants: (**A**) by U tube, (**B**) by semisolid agar plate, (**C**) by electron microscopy. The experiment was repeated in triplicate, and the results were consistent.

## Data Availability

Not applicable.

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
