# Peer review of "The Loss of focA Gene Increases the Ability of Salmonella Enteritidis to Exit from Macrophages and Boosts Early Extraintestinal Spread for Systemic Infection in a Mouse Model"

_microorganisms, 2022, doi:10.3390/microorganisms10081557_

Round 1

Reviewer 1 Report

In the manuscript entitled “The loss of focA gene increases the ability of Salmonella Enteritidis to exit from macrophages and boosts extra-internal spread for systemic infection in the mouse model” the authors characterized the role/s of focA gene in vivo. Below are the suggestions to improve the manuscript.

1.       Did the authors overexpress focA gene in the SEΔfocA mutant background? What would be the expected phenotype? The authors should discuss this possibility.

2.       Did the authors check the expression of other reported genes such as sifA, prgJ in SEΔfocA mutant? This way they would have established a link between focA and macrophage escape. The authors should discuss this aspect.

3.       What are the expression levels of Salmonella pathogenicity island genes in SEΔfocA mutant? This way the authors would have tried to establish a mechanistic pathway for the mutant phenotype. The authors should discuss this aspect.

4.       How the SEΔfocA mutant can cause higher cytotoxicity? The authors should discuss.

 The figure legends should mention how many times the experiments were conducted.

Author Response

Dear reviewer:

        First, I would like to thank the editors and reviewers for offering these helpful comments and good suggestions. I have carefully responded to these comments and revised this manuscript according to the comments made and the manuscript has been greatly improved by a senior expert. The answers are listed one by one.

Q1: Did the authors overexpress focA gene in the SEΔfocA mutant background? What would be the expected phenotype? The authors should discuss this possibility.

A: In this study, a focA deletion strain was constructed and a complementary strain was also constructed by high copy expressing vector, pBR322-focA. The phenotype of the complemented strain was almost consistent with that of the parental phenotype. We have discussed in the part of discussion.

Q2: Did the authors check the expression of other reported genes such as sifA, prgJ in SEΔfocA mutant? This way they would have established a link between focA and macrophage escape. The authors should discuss this aspect.

A: See response after Q3.

Q3: What are the expression levels of Salmonella pathogenicity island genes in SEΔfocA mutant? This way the authors would have tried to establish a mechanistic pathway for the mutant phenotype. The authors should discuss this aspect.

A: Thanks for the reviewer’s kind suggestions. We are still conducting further research on the mechanism of Salmonella extracellular escape. Questions 2 and 3 are relevant, so we will answer them together. We are also trying to find the mechanism by focusing on focA. We added them in discussion. At the first, SifA is SPI-2 protein, is crucial to the integrity of SCV. It was reported that when SCV is destroyed, Salmonella has the opportunity to escape into the cytoplasm. PrgJ is SPI-1 protein which expressed PrgJ protein to activated intracellular inflammasome to promote Salmonella exit from cells. Thus, sifA may be decreased and prgJ will be increased the expression in SEΔfocA. Second, the SEΔfocA induces no obvious change in cellular pyroptosis and apoptosis based on Caspase-1 protein measurement by SEΔfocA. Apoptosis and pyrosis are related to SPI-1. The expression levels of Salmonella pathogenicity island genes in SEΔfocA mutant will continue to be studied in our ongoing research. We have discussed this in the section of discussion.

Q4: How the SEΔfocA mutant can cause higher cytotoxicity? The authors should discuss.

A: SEΔfocA mutant increases the ability of Salmonella Enteritidis to exit from macrophages and boosts extra-internal spread for systemic infection, thus it may be one of the causes of high cytotoxicity .

Meanwhile, we learned the FocA protein functions as a bidirectional channel in enterobacteria which translocates harmful formate metabolites across the cytomembrane and contributes to its survival under anaerobic conditions. It had reported that formate is an intracellular signal that modulates virulence in response to bacterial metabolism. It was very interesting in our study showed that the formate expression of SEΔfocA was lower in both cytosol and bacteria. Therefore,the increase of cytotoxicity caused by SEΔfocA mutant is likely to be caused by many factors. It is worth for us continuous exploration. We have discussed this in the section of discussion. 

Q5: The figure legends should mention how many times the experiments were conducted.

A: All experiments in this paper were carried out three times. Data are presented as mean ± SEM of three independent experiments, We have added this sentence in the section of every figure legends.

       Please see the attachment for the revised version.

Best regards

Shizhong Geng.

Reviewer 2 Report

The paper is well written and clear and concise. The authors have identified an important factor in the interaction of Salmonella and macrophages in focA. They have identified focA as an important player in Salmonella exit from macrophages and subsequent spread.

The experimental design and relevant data are presented appropriately. The conclusions made are supported by the data presented.

There are minor English/grammar corrections needed throughout the manuscript.

Author Response

Dear reviewer:
        First,  I would like to thank you for your good suggestion.

       We have carefully revised the part about English and grammatical errors in the article.

      Please see the attachment for the revised version.

Best regards

Shizhong Geng.

Round 2

Reviewer 1 Report

The authors have addressed the comments satisfactorily.